# Postnatal quality of care measures for mothers and newborns at home: A scoping review

Ann-Sofie Mespreuve[1‡]*, Lise Apers[1‡], Ann-Beth Moller[2], Anna Galle[1]

**1** Department of Public Health and Primary Care, International Centre for Reproductive Health, Ghent University, Ghent, Belgium, **2** School of Public Health and Community Medicine, Institute of Medicine, Sahlgrenska Academy, University of Gothenburg, Gothenburg, Sweden

‡ A-SM and LA share first authorship on this work.
* annsofie.mespreuve@hotmail.com

**Data Availability Statement:** The data extraction for this scoping review can be found in Supplementary: S2 Table. Data Extraction Form.

**Funding:** The authors received no specific funding for this work.

## Abstract

The postnatal period is one of the most critical periods in the lives of mothers and newborns. Yet, the postnatal period remains the most neglected period along the maternal health care continuum. Globally, measures assessing quality of postnatal care (PNC) often focus on care at health facility level, the provision of home-based PNC and associated quality of care measures seem largely overlooked. This scoping review aims to give an overview of the literature on measures assessing quality of PNC for mothers and newborns in a home-based setting. This review was conducted according to the Arksey and O'Malley's methodology for scoping reviews. Three electronic bibliographic databases were searched together with a grey literature search. Two reviewers independently screened the identified articles. All data on home-based PNC measures were extracted and mapped according to the 2022 World Health Organization PNC Guideline recommendations in three categories: i) maternal care, ii) newborn care, iii) health system and health promotion interventions. Several additional quality of care domains, characterizing home-based PNC, were identified: i) social and emotional empowerment, ii) assessment of the home setting, iii) early breastfeeding, iv) health education and counseling, v) personal hygiene and prevention of infections, vi) referral to health facility when necessary, vii) thermal care, viii) parent-child relationship and ix) promote economic self-sufficiency. This review illustrates that home-based PNC has a very broad spectrum of care and plays a vital role in improving maternal and newborn health and well-being. In addition, there is a clear need for more research on the optimal timing and content of home-based care in the postnatal period for maximizing its potential.

## Introduction

Nearly 800 women worldwide died every day in 2020 from causes associated with pregnancy and childbirth according to latest estimates [1]. In addition, approximately 2.3 million newborn deaths were reported in 2021, representing 46% of all deaths under the age of five [2].

**Competing interests:** The authors have declared that no competing interests exist.

Most deaths are preventable by providing quality evidence-based care during and after childbirth, including support and careful monitoring in the postnatal period [3]. Postnatal care (PNC) is defined as the care given to mothers and their newborns in the first six weeks after childbirth [4].

Although the postnatal period is one of the most critical periods in the lives of mothers and newborns, it remains often the most neglected phase in maternal and childcare provision [5]. In 2022, the World Health Organization (WHO) published the "WHO recommendations on maternal and newborn care for a positive postnatal experience" with the aim to enhance the quality of routine PNC [4]. The WHO also refers to the "Quality of Care Framework" which places provision and experience of care at the core [4]. This framework emphasizes the importance of a positive experience as part of care and well-being for mothers and newborns [4]. PNC is a continuation of medical and non-medical care that the woman receives and is specified to meet mothers', newborns' and families' individual needs and preferences [6]. It entails physical checkups of both mother and child, attention to assessing the mental well-being of the mother and family, as well as family planning counselling [6]. In the literature, the majority of measures focus on facility-based care which reveals a gap in PNC measures after hospital discharge [7].

Over the last decade there is an international trend to shorten the length of stay in hospitals after birth, mainly to reduce costs [8, 9]. Therefore, utilization of outpatient health services or home visits by trained health professionals are becoming increasingly important to ensure quality PNC [10]. PNC at home includes professional care, self-care and family care, contributing to maternal and newborn health and well-being [7]. It has been reported that home-based postpartum care is linked with reduced neonatal mortality, increased rates of exclusive breastfeeding, and is cost-effective in enhancing newborn health outcomes [11].Some components characterizing PNC in the home-based context include assessment of the home environment, social and emotional support, and empowerment of mothers and their families [4,12–15].

In this regard, it is important to have an understanding of what measures are currently available examining PNC quality at home. This scoping review aims to provide an overview of the existing home-based PNC measures documented in peer-reviewed articles and grey literature in order to explore the care components and the associated quality measures. The focus of this review lays on the provision of care rather than the experience of care.

## Methods

### Study design

We conducted a scoping review according to Arksey and O'Malley's methodology for scoping reviews [16]. This method includes 5 steps: 1. identifying the research question, 2. identifying relevant studies, 3. study selection, 4. charting the data, 5. collating, summarizing and reporting the results. The population/concept/context (PCC) framework by the Joanna Briggs Institute, recommended for scoping reviews, was used to identify the main concepts of the research question and to inform the search strategy [17]. Inclusion criteria for identifying eligible studies are described in Table 1. The study protocol was registered in the Open Science Framework (DOI 10.17605/OSF.IO/APF5M) [18]. Results are presented according to the Preferred Reporting Items for Systematic reviews and Meta-Analyses extension for Scoping Reviews (PRISMA-ScR) Checklist [19] (S1 Fig).

Based on the PCC framework, the following eligibility criteria have been articulated:

- Geographical area: since the aim is to assess care provision on a global level, studies in all countries worldwide were eligible.

**Table 1. Inclusion criteria for identifying eligible studies.**

| PCC element | Description |
|---|---|
| Population | Mothers, their partners, newborns, parents/caregivers and families in the period after birth until 6 weeks postpartum |
| Concept | Home based postnatal care, which is defined as all care provided to mothers, their partners, newborns, parents/caregivers and families during the six weeks after childbirth at home. This involves medical care, but also non-medical assistance, including support in household, hygiene and psycho-social wellbeing. |
| Context | All contexts |

- Language: peer-reviewed research papers and grey literature in English.

- Publication date: articles with publication dates from 2010 to 2023 were included, as updated technical guidelines on postpartum and postnatal care were published by WHO in 2010 [20].

- Study types: qualitative, quantitative and mixed method studies. Both original research articles and review articles. Relevant grey literature such as policy documents and guidelines.

- Information sources: data were included when the topic covered any quality of care measure related to the provision of PNC in a home-based setting. Therefore, sources that focused exclusively on the experience of care or on provision of care solely possible in a facility-based setting were excluded. Moreover, articles have been included that cover any care component of home-based PNC, which has a broader scope than just recommended measures or practices. While validity and utility studies were considered, they were not the main focus of this scoping review.

## Search strategy

The scoping review included studies from the 1st of January 2010 to the 31st of October 2023. Search strings were developed for three databases namely PubMed, Embase and Web of Science. For PubMed, MeSH-terms were retrieved for the important concepts, including "postnatal care" and "home care services". The full search strings of all databases, which can be found as an additional file (S1 Table), were kept rather broad to make sure none of the relevant articles were missed. Additionally, grey literature was searched through the Google search engine and by consulting the websites of the WHO, United Nations (UN) iLibrary and the Demographic and Health Survey (DHS) [21–23]. The key words used for these searches were "postnatal care" and "home-based".

## Study selection

All acquired articles were saved in Mendeley and imported to Rayyan software for the screening process. Selection and screening of the acquired articles were conducted by two researchers (AM and LA) independently of each other and inconsistencies were discussed among the researchers. This implies arguments were presented and discussed until consensus was reached. Firstly, all duplicates were searched manually and removed. Secondly, articles that were not original, not written in English or that were published before 2010 were removed. Afterwards, articles were screened based on title and abstract and/or executive summaries. When the information found was not satisfactory enough to decide upon inclusion, full articles were read. All articles that fulfill the eligibility criteria were further screened. Additionally, a

grey literature search was conducted to make sure relevant guidelines and policy documents were identified.

## Data extraction and analysis

A data extraction form was developed in Microsoft Excel and piloted with the first 10 articles. After piloting, the form was adapted according to feedback of the research team (AM, LA and AG). The final data extraction form included authors, title, publication year, source, purpose of study, study design, geographical location of the study, home-based PNC measures and whether the measures were clearly defined, number of PNC components, timing and frequency of home visits, quality of care measures, funding resources and declared conflicted interests. The completed form can be found as an additional file (S2 Table). Articles were considered 'clearly defined' if the measures were sufficiently described to be reproducible [24]. Data extraction was done by two researchers independently (AM and LA) and conflicting results were discussed until agreement was reached. Subsequently measures were mapped according to the 2022 WHO PNC Guideline to give an overview of the available quality of care measures for home-based PNC. Quality of care measures that could not be linked to a specific recommendation were categorized in thematic areas ("quality of care domains") and listed in a separate table. A distinction was made between measures intended exclusively for post-institutional births and PNC measures following home births. When the articles did not specify, it was assumed that the measures applied to both institutional and home births.

## Results

The bibliographic database searches initially identified 262 results, of which 66 duplicates were removed. This resulted in 196 records to be screened. After removing the records that did not fulfill the eligibility criteria, 47 records were retained to be screened based on full text. Additionally, grey literature search yielded 12 eligible records. In total, 59 articles were full text screened and 31 records were included in the review. Detailed flowchart based on PRISMA guidelines is outlined in Fig 1 [25].

Most articles focused on multiple regions (n = 10; 32%), followed by the Region of the Americas (n = 8; 26%), and the South-East Region (13%). Three studies took place in the European Region (10%), three in the Eastern Mediterranean Region (10%), only two in the African Region (6%) and only one in the Western Pacific Region (3%). The most frequently used study designs were cross-sectional studies (n = 9; 29%) and international guideline documents (n = 9; 29%). The second-most used study designs were (quasi) experimental studies (n = 8; 26%). Two studies were validation studies (6.5%), two were systematic reviews (6.5%), and one was a longitudinal study (3%). Most of the included studies did not have clearly defined measures (n = 22; 71%). However, nine studies had a clear definition of their used measurements of quality of care (29%). Most of the studies relied on data collected through observations (n = 15; 48%) and five used a combination of data sources (16%). Only one study used health program records (3%) and another one used population-based household data (3%). For nine studies, the data source of the original study was either not applicable (e.g. guideline documents) or not defined by the authors (29%). An overview of these results can be found in Table 2.

## Timing and frequency of home-based PNC visits

Both timing and frequency differed across different contexts. Within experimental studies, the frequency of home visits varied according to the study design, ranging from one to 28 visits.

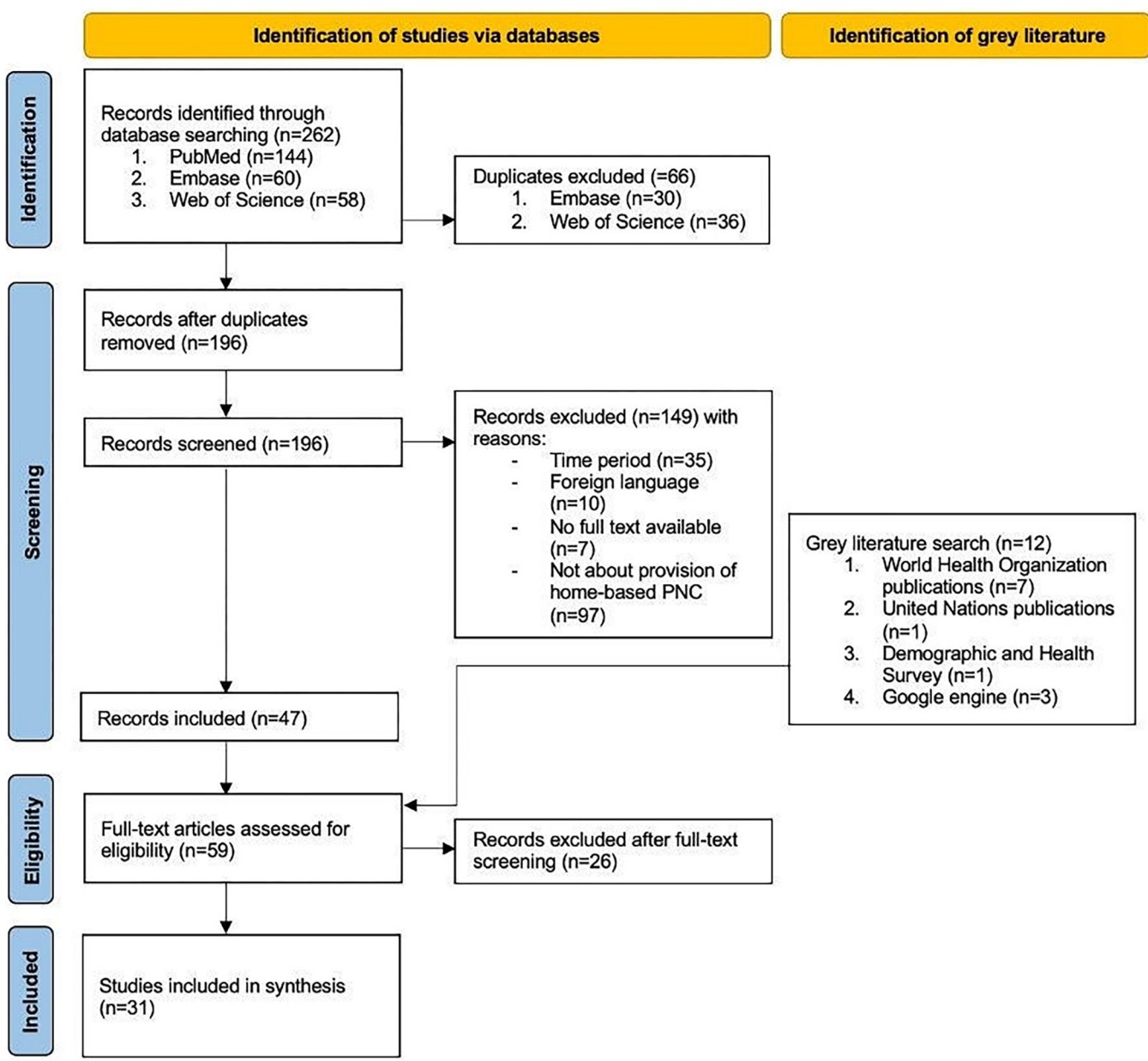

**Fig 1. Flowchart of included studies [25].**

According to WHO recommendations, three to five PNC contacts are advised for facility-based deliveries, with an additional contact recommended for home births.

With regard to timing, experimental studies display a wide range of intervals. A comprehensive overview of these timings is provided in S2 Table. The WHO recommendations have remained relatively consistent over the years. The most recent guidelines suggest scheduling a first visit between 48 and 72 hours postpartum. If the mother gave birth at home, a visit is recommended as early as possible within the first 24 hours postpartum. Finally, PNC should be offered until the 6th week postpartum according to the recommendations.

**Table 2. Characteristics of the included studies.**

| WHO Region* | Number of studies in % and (number of studies) |
|---|---|
| African Region | 6 (n = 2) |
| Region of the Americas | 26 (n = 8) |
| South-East Region | 13 (n = 4) |
| European Region | 10 (n = 3) |
| Western Pacific Region | 3 (n = 1) |
| Eastern Mediterranean Region | 10 (n = 3) |
| Multi region | 32 (n = 10) |
| **Study design** | |
| Cross-sectional | 29 (n = 9) |
| Longitudinal | 3 (n = 1) |
| (Quasi) experimental | 26 (n = 8) |
| Guideline document | 29 (n = 9) |
| Validation study | 6.5 (n = 2) |
| Systematic review | 6.5 (n = 2) |
| **Clearly defined measures** | |
| Yes | 29 (n = 9) |
| No | 71 (n = 22) |
| **Data source(s) of the original studies** | |
| Population based household data | 3 (n = 1) |
| Health program records | 3 (n = 1) |
| Not defined/applicable | 29 (n = 9) |
| Observations | 48 (n = 15) |
| Combination of sources | 16 (n = 5) |

* https://www.who.int/about/who-we-are/regional-offices.

**Provision of care measures–Mothers.** For providing maternal care in a home-based setting in the postnatal period, the literature reported measures for six out of the 24 recommendations of the WHO guideline (see Table 3). Most measures were found for the recommendation on physiological assessment of the mother, with 11 different sources mentioning this. Moreover, for the recommendations on screening for postpartum depression and anxiety, and postpartum contraception, seven sources described measures. In addition, measures regarding recommendations on prevention of postpartum depression and anxiety, postpartum oral iron and folate supplementation, and physical activity, were discussed in one, three and two sources, respectively. Remarkably, for the category of preventive measures no home-based measures could be identified in the literature.

## Provision of care measures–Newborns

For providing newborn care in a home-based setting in the postnatal period, the literature reported measures for ten out of the 19 numbered recommendations of the WHO guideline (see Table 4). The recommendation with the most reported measures was on assessment of the newborn for danger signs, with 16 different resources mentioning this. Moreover, 12 resources described measures regarding the guideline on responsive care to infants and children 0 and 3 years of age, and those were covering early childhood development and breastfeeding support. In addition, measures were mentioned for the following recommendations: universal screening for abnormalities of the eye, universal screening for neonatal hyperbilirubinemia, timing

**Table 3. Identified measures on provision of maternal care as per the WHO PNC guideline recommendations for a positive postnatal experience.**

| Number of recommendation | Maternal assessment | References | Identified measures |
|---|---|---|---|
| 1 | Physiological assessment of the woman | [12] [13] [26] [27] [28] [29] [30] [31] [32] [4] [14] [5] | ▪ Assessment of parent health [12]<br>▪ Maternal observations: maternal vital signs (including pulse, temperature and blood pressure); postnatal observation of lochia, breasts, urinary function, bowels, perineum, and caesarean wound) [13]<br>▪ Assessment of maternal health [26]<br>▪ Counseling on early identification of postpartum complications after home birth (including heavy bleeding, severe pain in abdomen, fever, convulsions or fits, foul smelling discharge, breast nipple problems) [27]*<br>▪ Examinations for lacerations and blood pressure [28]<br>▪ Observing blood flow, measuring temperature [29]*<br>▪ Examinations of oral and dental health risk factors, common complaints in the postpartum period [30]<br>▪ Assessing vital signs, consciousness, convulsion, breathing problems, defection problems, vertigo, inflammation of the gums, shock, comorbidities and medical history [31]*<br>▪ Examination of extremities, breasts, eyes, abdomen and urinary and reproductive organs [31]*<br>▪ Physical assessment of mother [32]<br>▪ Assessment of the woman's physical well-being [4]*<br>▪ Assessment of the mother [14]*<br>▪ Assessing danger signs [5]* |
| | **Mental health interventions** | | |
| 18 | Screening for postpartum depression and anxiety | [12] [13] [30] [31] [32] [4] [14] | ▪ Assessment of maternal depression and anxiety [12]<br>▪ Measurement of postnatal depression with a psychological/social/behavioral instrument [13]<br>▪ Examinations and observation of mental, and psychological health (Edinburgh Postnatal Depression questionnaire to screen for postpartum depression) [30]<br>▪ Assessing symptoms of psychological disorders [31]*<br>▪ Screening for postpartum depression [32]<br>▪ Assessment of the woman's emotional well-being [4]*<br>• Assessment of maternal mental health [14]* |
| 19 | Prevention of postpartum depression and anxiety | [33] | • Emotional and cognitive support: strategies for decreasing anxiety, means for strengthening coping methods, and supportive decision-making [33] |
| | **Nutritional interventions and physical activity** | | |
| 20 | Postpartum oral iron and folate supplementation | [29] [30] [34] | ▪ Providing iron-folic acid supplementation [29]*<br>▪ Examinations, observations, questions and necessary instructions and training with regard to nutrition in the postnatal period and use of supplements [30]<br>▪ Provide supplements for lactating mothers [34]*<br>▪ Ensuring nutrient intake in meals and intake of micronutrient supplements [34]* |
| 22 | Physical activity | [30] [35] | ▪ Examinations, observations, questions and necessary instructions and training with regard to exercise activities [30]<br>▪ Physical health: exercise [35]* |
| | **Contraception** | | |
| 24 | Postpartum contraception | [27] [36] [29] [30] [31] [35] [37] | ▪ Counseling of the couples to choose an appropriate family planning method, counseling on contraception [27]*<br>▪ Home-based contraceptive delivery [36]*<br>▪ Providing contraceptive methods [29]*<br>▪ Examinations, observations, questions and necessary instructions and training with regard to sexual health and contraception [30]<br>▪ Consultation on family planning [31]*<br>▪ Life course: family planning [35]*<br>▪ Family planning [37]* |

Measures marked with * pertain to studies that also include home-births.

**Table 4. Identified measures on provision of newborn care as per the WHO PNC Guideline recommendations for a positive postnatal experience.**

| Number of recommendation | Newborn assessment | References | Identified measures |
|---|---|---|---|
| 25 | Assessment of the newborn for danger signs | [33]<br>[12]<br>[13]<br>[38]<br>[26]<br>[27]<br>[28]<br>[29]<br>[32]<br>[39]<br>[5]<br>[40]<br>[41]<br>[42]<br>[4]<br>[14]<br>[38]<br>[43]<br>[14] | ▪ Infant physical evaluation: At each follow-up, infants were physically examined including their body temperature, pulse rate, respiration rate, blood pressure, peripheral saturation, height, body weight, and head circumference [33]<br>▪ Assessment of infant health [12]<br>▪ Neonatal observations: observation of bowel function, urine output, feeding and skin color; infant weighing [13]<br>▪ Identification of signs of severe neonatal morbidities and referral to a health facility [38]*<br>▪ Assessment of infant health [26]<br>▪ Counseling on danger signs in newborn and referring appropriately [27]*<br>▪ Infants' weight [28]<br>▪ Identifying conditions/danger signs requiring additional care [29]*<br>▪ Counseling on when to take a newborn to a health facility [29]*<br>▪ Physical assessment of the infant [32]<br>▪ Timely recognition of danger signs [39]*<br>▪ Treatment and referral (to health facility services) when needed [39]*<br>▪ Assessing danger signs and referral to a health facility when needed [5]*<br>▪ Check-up of clinical signs of severe illnesses: stopped feeding well, history of convulsions, fast breathing, severe chest in-drawing, no spontaneous movement, fever, low body temperature, any jaundice in first 24 hours of life [40]*<br>▪ Assess for danger signs: not able to feed or stopped feeding well, convulsions, fast breathing, chest indrawing, high or very low temperature, movement, local infection, measure birth weight and identify small babies [41]*<br>▪ Assess a newborn for danger signs [42]*<br>▪ Measure weight to identify small baby [42]*<br>▪ Identify when a newborn needs referral and assist the family in going to a health facility [42]*<br>▪ Assessment of the newborn's physical well-being [4]*<br>▪ Assessment of the newborn [14]*<br>▪ Identification of signs of severe neonatal morbidities and referral to a health facility [38]<br>▪ Facilitated referral for newborn illness [43]*<br>• If necessary, referral to other health professionals or agencies [14] |
| 26 | Universal screening for abnormalities of the eye | [27]<br>[44] | ▪ Counseling on care of the eye of a newborn [27]*<br>▪ Eye care [44]* |
| 28 and 29 | Universal screening for neonatal hyperbilirubinemia | [40]<br>[41] | ▪ Check-up of yellow palms and soles at any age [40]*<br>▪ Asses for danger signs: yellow soles [41]* |
| | **Preventive measures** | | |
| 30 | Timing of first bath to prevent hypothermia and its sequelae | [44]<br>[45]<br>[40]<br>[14] | ▪ Delayed bathing [44]*<br>▪ First bathed after 72 hours or more following birth [45]*<br>▪ Reinforce delayed bathing [40]*<br>▪ Delayed bathing [14]* |
| 31 | Use of emollients for the prevention of skin conditions | [33]<br>[29]<br>[14] | ▪ Basic care skills: skin care [33]<br>▪ Providing skin care [29]*<br>▪ Attention to infant skin care [14]* |
| 32 | Umbilical cord care | [33]<br>[13]<br>[38]<br>[27]<br>[29]<br>[45]<br>[44]<br>[39]<br>[40]<br>[14] | ▪ Care for umbilical cord [33]<br>▪ Routinely check the umbilical cord [13]<br>▪ Health education and counseling of families regarding hygienic cord care [38]*<br>▪ Counseling on care of cord [27]*<br>▪ Providing hygienic umbilical cord care [29]*<br>▪ Complete cord care after non-institutional births and check-up whether the respondent used any modern instrument (blade or scissors) to cut the cord, whether the instrument was boiled before use, and whether nothing was applied to the cord [45]*<br>▪ Cord care [44]*<br>▪ Support for hygienic cord care [39]*<br>▪ Cord care [40]*<br>• Attention to umbilical cord care [14]* |
| 34 | Immunization for the prevention of infections | [40] | • Reinforce immunization [40]* |

*(Continued)*

**Table 4.** (Continued)

| Number of recommendation | Newborn assessment | References | Identified measures |
|---|---|---|---|
| **38** | Responsive care to infants and children 0 and 3 years of age | [46]<br>[47]<br>[13]<br>[48]<br>[30]<br>[31]<br>[35]<br>[32]<br>[41]<br>[42]<br>[4]<br>[14] | Early childhood development:<br>▪ Optimize child development [48]*<br>▪ Early childhood development: temperament, development (social/physical), appropriate expectations [35]*<br>Breastfeeding support<br>▪ Breastfeeding education/support at home based on the WHO/UNICEF breastfeeding counseling/lactation management courses [46]<br>▪ Breastfeeding training according to the plan [47]<br>▪ Breast examination in terms of problems and breastfeeding [47]<br>▪ Observing breastfeeding and discovering problems [47]<br>▪ Emphasizing breastfeeding according to the desire and demand of the infant [47]<br>▪ Breastfeeding assessment and support [13]<br>▪ Examinations, observations, questions and necessary instructions and training with regard to breastfeeding with its related problems and duration [30]<br>▪ Consultation on breastfeeding [31]*<br>▪ Addressing breastfeeding [32]<br>▪ Support breastfeeding: initiation, attachment and positioning [41]*<br>▪ Support the mother to initiate and sustain breastfeeding: assess attachment and suckling and help her to improve position and attachment [42]*<br>▪ Breastfeeding promotion and support [4]*<br>▪ Education on breastfeeding [14]* |
| | **Breastfeeding** | | |
| 42 | Exclusive breastfeeding | [47]<br>[38]<br>[44]<br>[39]<br>[40]<br>[41]<br>[42]<br>[14] | ▪ Emphasizing exclusive breastfeeding [47]<br>▪ Health education and counseling on exclusive breastfeeding [38]*<br>▪ Exclusive breastfeeding [44]*<br>▪ Support exclusive breastfeeding [39]*<br>▪ Continue to promote exclusive breastfeeding [40]*<br>▪ Support the mother to sustain exclusive breastfeeding [41]*<br>▪ Advise families on exclusive breastfeeding [42]*<br>▪ Attention to exclusive breastfeeding [14]* |

Measures marked with * pertain to studies that also include home-births.

of first bath to prevent hypothermia and its sequelae, use of emollients for the prevention of skin conditions, umbilical cord care, immunization for the prevention of infections, and exclusive breastfeeding. The number of resources varied between one and ten.

## Health systems and health promotion Interventions

We found existing measures for two of the 12 recommendations (see Table 5). Five resources provided a measure on the guideline of men's involvement in PNC. Only two resources mentioned the use of home-based records.

## Unique quality of care measures, characterizing home-based PNC in the literature

We have found a broad range of home-based PNC measures in the literature that could not be mapped under the recommendations of the 2022 WHO PNC Guideline and that seem to characterize home-based PNC. They covered topics such as social and emotional support and empowerment, assessment of the home environment and context, early breastfeeding, health education and counseling, personal hygiene and prevention of infections, referral to a health facility when necessary, thermal care, promotion of parent-child relationship and promotion

**Table 5. Identified measures for health systems and health promotion interventions as per the WHO Guideline recommendations for a positive postnatal experience.**

| Number of recommendation | Health systems and health promotion interventions | References | Identified measures |
|---|---|---|---|
| 52 | Men's Involvement in PNC | [35] [4] | ▪ Caregiver support: father involvement [35]* ▪ Involvement of men in PNC and maternal and newborn health [4]* |
| 53 | Home-based records | [20] [4] | ▪ Maintain a home-based maternal and child health record [20] ▪ Use of home-based records [4]* |

Measures marked with * pertain to studies that also include home-births.

of economic self-sufficiency (see Fig 2). An overview of these additional quality of care measures can be found in Table 6.

## Discussion

We found a wide range of studies examining PNC quality in a home-based setting, providing a wide array of quality of care measures. The frequency, content and measures of home-based PNC differed across the different contexts. Notably, the majority of measures discussed in this review were described vaguely, suggesting that the published information is not comprehensive enough to be reproducible. Moreover, it was noted that quality of care indicators, such as person-centered care and respectful care, were discussed limitedly in the included studies. Comparing our results with another recent review on postnatal quality of care measures (not focused on a home-based setting) [7], we found that in addition to the clinical and biomedical

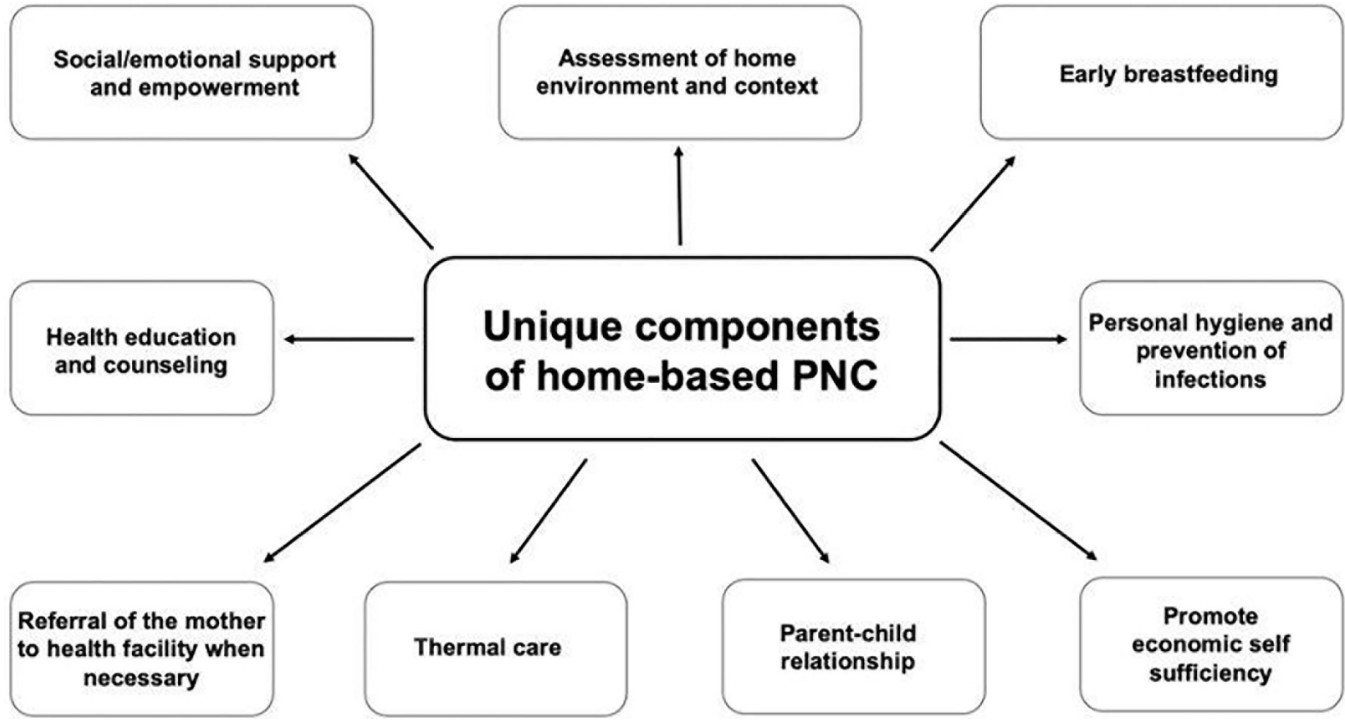

**Fig 2. Overview of the unique quality of care domains, characterising home-based PNC.** Figure manually created for this study.

aspects of PNC, care at home is also oriented towards social and emotional empowerment, assessment of the home setting, and stimulating the parent-child relationship.

In the postnatal period support is crucial, from both health professionals and family members [50–52]. Our review shows that an important aspect of home-based PNC is providing social and emotional support for the whole family (including mother, newborn, father and broader family), in addition to the biomedical aspects of maternal and newborn care. In that aspect, PNC at home is filling a critical gap in care. Studies have shown that poor levels of social support and economic concerns are risk factors for poor perinatal mental health, which

**Table 6. Identified measures on provision of newborn and maternal care that are not linked to specific WHO PNC Guideline recommendations for a positive postnatal experience.**

| Quality of care domain | References | Identified measures |
|---|---|---|
| Social/emotional support and empowerment | [33]<br>[15]<br>[12]<br>[47]<br>[13]<br>[26]<br>[28]<br>[31]<br>[35]<br>[32]<br>[37]<br>[4]<br>[14]<br>[49] | ▪ Improving participation of the family in care, and supporting caregivers [33]<br>▪ Emotional and cognitive support: mothers were provided information on baby care, strategies for decreasing anxiety, means for strengthening coping methods, and supportive decision-making [33]<br>▪ Supporting mothers: presence, listening, protection, and translation between parents' expectations and the challenges of integrating a new infant into the family [15]*<br>▪ Empowering mothers: sharing information and affirming mothers' values and opinions so that their transition to motherhood was optimal [15]*<br>▪ Empowering families: flexibility, following family cues, and assisting families to develop parenting skills [15]*<br>▪ Parental social support from others [12]<br>▪ Creating and strengthening self-esteem and emotional support for mothers [47]<br>▪ Assessment of social support [13]<br>▪ Provision of ongoing education to mother, partner and other family members [13]<br>▪ Assessment of mental health and social-emotional support [26]<br>▪ Emotional and informational family support for both parents [28]<br>▪ Greeting and establishing an intimate relationship with the mother [31]*<br>▪ Emotional health: maternal mental health, stress, coping, well-being [35]*<br>▪ Relationships: communication, relationship with parents [35]*<br>▪ Addressing social connectedness [32]<br>▪ Couple relationship/communication exercises [37]*<br>▪ Involvement of men in PNC and maternal and newborn health [4]*<br>▪ Emotional or practical support (to families) [14]*<br>▪ Triadic interactions that involve child, parent and home visitor: observing, modeling, coaching → better parent-child relationship → improving child development outcomes [49]* |
| Assessment of home environment and context | [12]<br>[13]<br>[26]<br>[48]<br>[31]<br>[35]<br>[32]<br>[20]<br>[4]<br>[14] | ▪ Assessment of medical home, family violence, mother's past experience of maltreatment, parental substance abuse [12]<br>▪ Assessment of the home environment [13]<br>▪ Assessment of substance abuse and family and community safety [26]<br>▪ Assessment of birthing parent history of parenting difficulties [26]<br>▪ Promote a nurturing home environment [48]*<br>▪ Identifying mother's socio-economic status and lifestyle [31]*<br>▪ Assessing alcoholism [31]*<br>▪ Identifying wife and any social abuse [31]*<br>▪ Physical health: substance use and smoking [35]*<br>▪ Relationships: domestic violence [35]*<br>▪ Child physical care: home safety [35]*<br>▪ Screening for substance use and intimate partner violence [32]<br>▪ Assess relevant safety issues for all family members in the home and environment [20]<br>▪ Assessment of the home environment [4]*<br>▪ Assessment of family circumstances and the home environment [14]* |
| Early breastfeeding | [38]<br>[45]<br>[44]<br>[39]<br>[40]<br>[14] | ▪ Early initiation of breastfeeding after home birth [38]*<br>▪ Early breastfeeding within 1 hour after home birth [45]*<br>▪ Feeding colostrum within 1h-6h after home birth [44]*<br>▪ Support to initiate early breastfeeding after home birth [39]*<br>▪ Continue to promote early breastfeeding after home birth [40]*<br>▪ Attention to early initiation of breastfeeding after home birth [14]* |

*(Continued)*

**Table 6.** (*Continued*)

| Quality of care domain | References | Identified measures |
|---|---|---|
| Health education and counseling | [33]<br>[12]<br>[13]<br>[38]<br>[26]<br>[48]<br>[27]<br>[43]<br>[29]<br>[30]<br>[31]<br>[35]<br>[32]<br>[34]<br>[37]<br>[20]<br>[42]<br>[4]<br>[14]<br>[5] | ▪ Basic care skills (e.g. changing diapers, diaper rash care, skin care, bathing, dressing, strengthening sleep etc.) were performed with mothers to promote appropriate skills [33]<br>▪ Assessment of childcare planning, parent-infant relationship, management of infant crying [12]<br>▪ Provision of ongoing education to mother, partner and other family members [13]<br>▪ Health education and/or counseling of families regarding neonatal care practices [38]*<br>▪ Education to improve caregiver recognition of life-threatening neonatal problems [38]*<br>▪ Education to improve health care-seeking behaviors [38]*<br>▪ Management of infant crying [26]<br>▪ Provide care coordination [48]*<br>▪ Counseling on diet, hygiene, rest and resumption of sexual intercourse [27]*<br>▪ Group-based health education [43]*<br>▪ Counseling provided during home-visits [43]*<br>▪ Counseling on birth spacing and nutrition [29]*<br>▪ Examinations, observations, questions and necessary instructions and training with regard to care for the infant and how to manage postpartum complications [30]<br>▪ Providing health education based on the woman's SES and health status [31]*<br>▪ Consultations on medicinal supplements [31]*<br>▪ Caregiver support: social/parent support, childcare, parenting classes [35]*<br>▪ Addressing unmet health needs [32]<br>▪ Home-based counseling [34]*<br>▪ Couple HIV counseling at home [37]*<br>▪ Promote safety education and use of basic safety equipment [20]<br>▪ Advise families on optimal care practices for the newborn [42]*<br>▪ Assist families to provide extra care for the small baby including frequent feeding [42]*<br>▪ Health education and counseling [4]*<br>▪ Health education and education on infant feeding [14]*<br>▪ Provide counselling and advice based on the mother's perception of her own health and that of her baby, on the health problems she had observed and her actions she had taken in case of symptoms, on her breastfeeding pattern and on what kind of social support she had at home [5]* |
| Personal hygiene and prevention of infections | [29]<br>[30]<br>[44]<br>[41]<br>[42]<br>[14] | ▪ Increasing hand washing [29]*<br>▪ Examinations, observations, questions and necessary instructions and training with regard to personal hygiene (of the mother) [30]<br>▪ Hand washing with soap and water [44]*<br>▪ Hand washing skills [41]*<br>▪ Advise families on using good hygiene to prevent infections [42]*<br>▪ Attention to hygiene (e.g. hand washing and water quality) [14]* |
| Referral of the mother to health facility when necessary | [13]<br>[48]<br>[35]<br>[34]<br>[39]<br>[42]<br>[14]<br>[5] | ▪ Referral to community support services [13]<br>▪ Link families to health services [48]*<br>▪ Information/referrals: emergency/crisis plan, housing, utilities [35]*<br>▪ Provide facility referral for testing and suspected complications throughout postpartum period [34]*<br>▪ Treatment and referral (to health facility services) when needed [39]*<br>▪ Identify when a woman needs referral and assist the family in going to a health facility [42]*<br>▪ If necessary, referral to other health professionals or agencies [14]*<br>▪ Referral to a health facility when needed [5]* |
| Thermal care | [38]<br>[29]<br>[45]<br>[39]<br>[42] | ▪ Care of the newborn immediately after birth: keeping the baby warm [38]*<br>▪ Keeping the baby warm [29]*<br>▪ Complete thermal protection: whether the baby was wiped and wrapped within <10 minutes after birth [45]*<br>▪ Thermal care: support for keeping the newborn warm [39]*<br>▪ Advise families on keeping the newborn warm [42]* |
| Parent-child relationship | [12]<br>[26]<br>[35]<br>[44]<br>[20]<br>[40]<br>[42]<br>[14] | ▪ Assessment of parent-infant relationship [12]<br>▪ Assessment of parent-infant relationship [26]<br>▪ Attachment, responsiveness, reciprocity, affection, empathy [35]*<br>▪ Kangaroo care [44]*<br>▪ Promote parent- or mother-to-child emotional attachment [20]<br>▪ Reinforce skin-to-skin contact [40]*<br>▪ Skin-to-skin care [42]*<br>▪ Skin-to-skin contact [14]* |
| Promote economic self sufficiency | [48] | • Promote economic self-sufficiency [48]* |

Measures marked with * pertain to studies that also include home-births.

sparked during the COVID 19 pandemic [53–57]. Also for violence detection and prevention (and potential referral) health providers visiting mothers at home play a vital role [35]. This review shows the rich content of home-based PNC and high potential for improving maternal and newborn health and well-being, going beyond the period in the hospital. Yet, most literature on PNC quality focuses on the clinical aspects of care in a hospital-based setting [58–60].

Several recommendations of the 2022 WHO PNC Guidelines were not covered in the included studies on home-based PNC (e.g. sleeping position for the prevention of sudden infant death syndrome and non-pharmacological interventions to prevent postpartum mastitis) [4]. Nevertheless, these interventions could be delivered in a home-setting and might highly contribute to improved health of mothers and newborns. The gaps in home-based care highlighted in this review might be a steppingstone to improve PNC programs and provide guidance on what care could and should be provided at home.

Quality of care measures for home-based PNC were very broad and definitions varied across studies. This scoping review identified several unique measures not included in the 2022 WHO PNC Guideline recommendations, such as social/emotional support and empowerment, assessment of home environment and context, and referral of the mothers to the health facility when necessary. These findings suggest that the WHO guidelines may not be sufficiently oriented towards home-based settings. From this review and the broader evidence, it is clear that the optimal package and content of home-based PNC are yet to be determined [61]. More evidence will be needed to standardize home-based PNC, together with specific guidelines and accompanied quality of care measures to track progress.

## Limitations

We recognize that this scoping review has several limitations. Firstly, only English-language literature was included, indicating that relevant studies in other languages might be missed. Secondly, the keywords "postnatal care" and "home-based" were used in our search strategy. However, some articles could contain important measures of home-based PNC but might not have referred to these keywords as home-based care can be described in different ways. Thirdly, publication bias should be taken into account. The balance of published research is in favor of positive results, indicating that certain research has never been published and thus, inaccessible. Moreover, attention was paid to capturing as many guidelines and policy documents as possible. However, since some documents, especially in low resource settings, might not be well-indexed in electronic databases and the researchers manually searched the web browser, it is possible that some eligible documents were not identified. Furthermore, we acknowledge that the classification of measures identified in the included articles as clearly defined or not is inherently subjective. A final limitation is that categorization of the identified measures into different unique subdomains of home-based PNC was subjective and intuitive, this should be interpreted as such and cannot be regarded as a validated framework or categorization.

## Future perspectives

We decided to focus on the provision aspect of the quality of care framework, therefore, further research is needed to unravel the experience of home-based PNC measures.

## Conclusion

This review identified certain key quality of care measures of home-based PNC, mainly around social and emotional empowerment, showing the vital role of home-based PNC in improving the well-being of mothers, newborns and their families. However, other important

components of PNC, mainly around preventive care, did not seem to be assessed in a home-based setting. In conclusion, more research will be needed to decide upon the ideal timing and content of home-based PNC, which can be translated into clear guidelines and accompanied quality of care measures, improving care for mothers, newborns and their families.

## Supporting information

**S1 Fig. Meta-analysis extension for Scoping Reviews (PRISMA-ScR) checklist.**
(DOCX)

**S1 Table. Search strategies (search terms and results).**
(PDF)

**S2 Table. Data extraction form.**
(PDF)

## Acknowledgments

We would like to thank the librarian of Ghent University, Ms. Nele Pauwels, for reviewing and improving the search strategy.

## Author Contributions

**Conceptualization:** Ann-Sofie Mespreuve, Lise Apers, Ann-Beth Moller, Anna Galle.

**Data curation:** Ann-Sofie Mespreuve, Lise Apers, Anna Galle.

**Formal analysis:** Ann-Sofie Mespreuve, Lise Apers, Ann-Beth Moller, Anna Galle.

**Methodology:** Ann-Sofie Mespreuve, Lise Apers, Anna Galle.

**Supervision:** Ann-Sofie Mespreuve, Lise Apers, Ann-Beth Moller, Anna Galle.

**Visualization:** Ann-Sofie Mespreuve, Lise Apers.

**Writing – original draft:** Ann-Sofie Mespreuve, Lise Apers.

**Writing – review & editing:** Ann-Beth Moller, Anna Galle.

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
