## [Decision Letter · Decision Letter 0]

19 Apr 2024

PGPH-D-24-00532

Postnatal quality of care measures for mothers and newborns at home: a scoping review

Dear Dr. Mespreuve,

Thank you for submitting your manuscript to PLOS Global Public Health. After careful consideration, we feel that it has merit but does not fully meet PLOS Global Public Health’s publication criteria as it currently stands. Therefore, we invite you to submit a revised version of the manuscript that addresses the points raised during the review process.

We look forward to receiving your revised manuscript.

Kind regards,

Hannah Tappis, DrPH, MPH

Academic Editor

Journal Requirements:

1. We ask that a manuscript source file is provided at Revision. Please upload your manuscript file as a .doc, .docx, .rtf or .tex.

2. Please provide separate figure files in .tif or .eps format only and remove any figures embedded in your manuscript file. Please also ensure all files are under our size limit of 10MB.

Additional Editor Comments (if provided):

Reviewers' comments:

Reviewer's Responses to Questions

**Comments to the Author**

1. Does this manuscript meet PLOS Global Public Health’s publication criteria? Is the manuscript technically sound, and do the data support the conclusions? The manuscript must describe methodologically and ethically rigorous research with conclusions that are appropriately drawn based on the data presented.

Reviewer #1: Partly

Reviewer #2: Yes

2. Has the statistical analysis been performed appropriately and rigorously?

Reviewer #1: N/A

Reviewer #2: N/A

3. Have the authors made all data underlying the findings in their manuscript fully available (please refer to the Data Availability Statement at the start of the manuscript PDF file)?

Reviewer #1: Yes

Reviewer #2: Yes

4. Is the manuscript presented in an intelligible fashion and written in standard English?

Reviewer #1: Yes

Reviewer #2: Yes

5. Review Comments to the Author

Reviewer #1: Thank you for giving me the opportunity to review this interesting article which takes a rigorous methodological approach to an important topic, there needs to be better clarity on the aims and a closer look at how the identified measures are reported/classified.

My major comments are around being clearer about the context and scope of the review.

1. The definition of the content of PNC starting in the introduction is very brief and should be expanded. Please reference the new guidelines here and specify some of the differences in facility and home-based PNC. This should be used to explain why different measures for facility and home PNC are needed i.e. is the content and approach different or is the difference more the location of the care?

2. Clarify if the review focuses on measures of recommended practices or on any measure of postnatal care? For example, does the review include measures used in studies testing new content or approaches?

3. In the introduction the authors talk about the growing importance of home visits for PNC, please back this up with coverage data and describe other strategies for PNC such as outpatient visits.

4. It would like to see a clearer research question that helps the reader understand the scope and focus of the review a little more. Is the aim to map out the core domains being measured or to say something about validity and utility?

5. My understanding is that the authors have mapped any measure that could have been proposed in international guidelines, used in a trial, used in a cross-section study measuring quality of existing services or that were the explicit focus of validation studies? Can this be clarified and included in the eligibility criteria and the descriptor of studies as it changes the utility of the review.

6. When they say measure are they referring to an indicator or also to items on a instrument that could be used for a composite measure?

7. It would be good to clarify the definition of quality used in the data extraction process. Specifically, where did they draw the line between coverage and quality measures? For example, would a measure of the % of women who reported counselling be included or would the measure have to be the % women who had a ‘quality” home visit?

8. In the method section please provide more details as to what a reproducible measure refers to. I found the reference cited too broad to understand how decisions on this were made.

A second major comment is around the identified measures and how they are described in the Tables.

1. The identified measures in the Tables are very vague. For example, what constituted assessment of maternal health? What is a measure to optimize child development. At the moment, these Tables are too cryptic and the measures need to be re-assessed, re-classified and edited for consistency.

2. When the authors say ‘Assessment of XYZ” is this a measure of whether an assessment occurred or not? Or something about the quality of the assessment? I found it hard to understand what was actually being measured from the Tables.

3. What is the difference between as assessment and an observation.

4. In Table 4 under assessment they have body weight and under observation infant weight. I was unclear what the difference was?

5. I was unclear why thermal care, early breastfeeding, parent-child relationships (and a few others) were included as “unique measures” as there are guidelines on these and they were included in other ways in the Tables.

6. I was unsure how data from health facility records (Table 2) were eligible – were these supervision record of home visitors or home visitor records collated and stored in the facility?

Some more specific comments are:

7. Under the unique measure section the authors specify that measures outside of the recommendations “characterize home based PNC”. Can they clarify what they mean by this? Are many countries adding additional items to home visits or is this because trials were included which had a wide range of content rather than the content of national programmes? Are these data biased by one or two countries?

8. I do not feel the conclusion that home PNC is oriented to social and emotional support is supported by the evidence in the results. First, there are 10 studies alone with umbilical cord care measures and 13 on social/emotional support. Plus, the fact that something is measured does not mean that care is more oriented to that area- it just reflects what is being measured/is measurable. Also, this is a scoping review and many studies are missed so I would not draw such strong conclusions.

9. As far as I can see there are no experience of care measures- is that correct?

10. The authors need to acknowledge the variation in contexts in the discussion.

Reviewer #2: Thanks for the chance to read and review this important manuscript. The paper is a valuable addition to the literature however could benefit from some improvements. I have provided detailed comments in the attached PDF for the authors' attention and response.

6. PLOS authors have the option to publish the peer review history of their article (what does this mean?). If published, this will include your full peer review and any attached files.

**Do you want your identity to be public for this peer review?** For information about this choice, including consent withdrawal, please see our Privacy Policy.

Reviewer #1: No

Reviewer #2: **Yes: **Aline Semaan

---

## [Decision Letter · Decision Letter 1]

19 Jun 2024

PGPH-D-24-00532R1

Postnatal quality of care measures for mothers and newborns at home: a scoping review

Dear Dr. Mespreuve,

Thank you for submitting your manuscript to PLOS Global Public Health. After careful consideration, we feel that while much of the previous reviewers' feedback was addressed, a few points of clarification and further reflection are required. Therefore, we invite you to submit a revised version of the manuscript that addresses the points raised during the review process.

We look forward to receiving your revised manuscript.

Kind regards,

Hannah Tappis, DrPH, MPH

Academic Editor

Journal Requirements:

1. Please amend your online Financial Disclosure statement. If you did not receive any funding for this study, please simply state: “The authors received no specific funding for this work.”

2. Please update your online Competing Interests statement. If you have no competing interests to declare, please state: “The authors have declared that no competing interests exist.”

3. Please upload your main article file as a .doc, .docx or .rtf file.

4. Please provide separate figure files in .tif or .eps format only and remove any figures embedded in your manuscript file. Please also ensure that all files are under our size limit of 10MB. You may leave the figure captions or legends in the manuscript.

Additional Editor Comments (if provided):

Reviewers' comments:

Reviewer's Responses to Questions

**Comments to the Author**

1. If the authors have adequately addressed your comments raised in a previous round of review and you feel that this manuscript is now acceptable for publication, you may indicate that here to bypass the “Comments to the Author” section, enter your conflict of interest statement in the “Confidential to Editor” section, and submit your "Accept" recommendation.

Reviewer #1: All comments have been addressed

Reviewer #2: (No Response)

2. Does this manuscript meet PLOS Global Public Health’s publication criteria? Is the manuscript technically sound, and do the data support the conclusions? The manuscript must describe methodologically and ethically rigorous research with conclusions that are appropriately drawn based on the data presented.

Reviewer #1: Yes

Reviewer #2: Yes

3. Has the statistical analysis been performed appropriately and rigorously?

Reviewer #1: N/A

Reviewer #2: N/A

4. Have the authors made all data underlying the findings in their manuscript fully available (please refer to the Data Availability Statement at the start of the manuscript PDF file)?

Reviewer #1: Yes

Reviewer #2: Yes

5. Is the manuscript presented in an intelligible fashion and written in standard English?

Reviewer #1: Yes

Reviewer #2: Yes

6. Review Comments to the Author

Reviewer #1: Authors have acted on my comments adequately.

Reviewer #2: Dear Authors

Thanks for revising the manuscript and responding the previously raised comments in such detail that helped clarify several grey areas in the paper. A few minor points remain for your consideration:

1- Th response letter clarifies that the focus of the paper is on the provision aspect rather than the experiences - this point should be mentioned clearly in the objectives; the definition of the concept in Table 1, and in the methods. Having the following sentence in the “information resources” section could otherwise be misleading to readers: “data were included when the topic covered any quality of care measure of PNC in a home-based setting”, particularly if studies included content on experiences of care were systematically excluded from the review (same as was applied to the study from Gaza in the response letter). Having said that, I remain uncertain about whether the lack of including results/papers on experiences of care was a choice of the authors or was a finding of the scoping review literature, as the response letter also includes this: “Moreover, it was noted that quality of care indicators, such as person-centered care and respectful care, were discussed limitedly in the included studies.” Making the lack of information on experiences of care a possible finding of the review. This distinction should be clearly clarified in the objectives and methods.

2- Limitations: “In addition, we decided to focus on the provision aspect of the quality of care framework, therefore, further research is needed to unravel the experience of home-based PNC measures.” if this was a choice of the authors then it falls beyond he scope of objectives and is not a limitation, but a recommendation for future research.

3- In the tables, the note defining the * is confusing because it refers to using the symbol when studies describe two different settings. It is difficult to distinguish which is which: “ Measures marked with * pertain to studies that include either exclusively home births or both facility-based births and home births”

4- The previous comment about breastfeeding support was to mean that in the table it currently falls under recommendation number 38 “responsive care to infants and children 0-3”, while there is recommendation n42 on breastfeeding which includes “Exclusive breastfeeding”.

5- In the first paragraph of the discussion, authors introduce new findings related to frequency of PNC home visits – which I fail to see in the results while this is clearly a finding. I recommend to add this to results’ section. Additionally, the information on timing of PNC are listed in S3 but not mention in the results. Moreover, it is not clear how the study design could influence the number of PNC visits just from this sentence alone: “Within individual studies, the frequency of home visits varied according to the study design, ranging from one to 28 visits”. is this related to experimental studies? This should be clarified and a distinction should be made between the results from the guideline documents and those related to experimental studies.

7. PLOS authors have the option to publish the peer review history of their article (what does this mean?). If published, this will include your full peer review and any attached files.

**Do you want your identity to be public for this peer review?** For information about this choice, including consent withdrawal, please see our Privacy Policy.

Reviewer #1: No

Reviewer #2: No

---

## [Editor Report · Decision Letter 2]

2 Jul 2024

Postnatal quality of care measures for mothers and newborns at home: a scoping review

PGPH-D-24-00532R2

Dear Ms. Mespreuve,

We are pleased to inform you that your manuscript 'Postnatal quality of care measures for mothers and newborns at home: a scoping review' has been provisionally accepted for publication in PLOS Global Public Health.

Best regards,

Hannah Tappis, DrPH, MPH

Academic Editor